# Mutation Analysis of Thin Basement Membrane Nephropathy

**DOI:** 10.3390/genes13101779

**Published:** 2022-10-02

**Authors:** Yosuke Hirabayashi, Kan Katayama, Mutsuki Mori, Hiroshi Matsuo, Mika Fujimoto, Kensuke Joh, Tomohiro Murata, Masaaki Ito, Kaoru Dohi

**Affiliations:** 1Department of Cardiology and Nephrology, Mie University Graduate School of Medicine, Tsu 514-8507, Japan; 2Department of Kidney Center, Suzuka Kaisei Hospital, Suzuka 513-8505, Japan; 3Department of Internal Medicine, Takeuchi Hospital, Tsu 514-0057, Japan; 4Department of Pathology, The Jikei University School of Medicine, Tokyo 105-0003, Japan

**Keywords:** autosomal dominant Alport syndrome, *COL4A3*, *COL4A4*, microhematuria, mutation, thin basement membrane nephropathy, variant

## Abstract

Thin basement membrane nephropathy (TBMN) is characterized by the observation of microhematuria and a thin glomerular basement membrane on kidney biopsy specimens. Its main cause is heterozygous mutations of *COL4A3* or *COL4A4*, which also cause late-onset focal segmental glomerulosclerosis (FSGS) or autosomal dominant Alport syndrome (ADAS). Thirteen TBMN cases were analyzed using Sanger sequencing, multiplex ligation-dependent probe amplification (MLPA), and exome sequencing. Ten heterozygous variants were detected in *COL4A3* or *COL4A4* in nine patients via Sanger sequencing, three of which were novel variants. The diagnostic rate of “likely pathogenic” or “pathogenic” under the American College of Medical Genetics and Genomics guidelines was 53.8% (7 out of 13 patients). There were eight single nucleotide variants, seven of which were glycine substitutions in the collagenous domain, one of which was a splice-site single nucleotide variant, and two of which were deletion variants. One patient had digenic variants in *COL4A3* and *COL4A4*. While MLPA analyses showed negative results, exome sequencing identified three heterozygous variants in causative genes of FSGS in four patients with no apparent variants on Sanger sequencing. Since patients with heterozygous mutations of *COL4A3* or *COL4A4* showed a wide spectrum of disease from TBMN to ADAS, careful follow-up will be necessary for these patients.

## 1. Introduction

Type IV collagen is a major constituent of the glomerular basement membrane (GBM), and a developmental isoform switch from triple-helical α1.α1.α2 (IV) to triple-helical α3.α4.α5 (IV) protomers occurs in the capillary loop stages [1]. Mutations in *COL4A3*, *COL4A4*, or *COL4A5*—encoding the α3, α4, or α5 (IV) chains, respectively—cause various diseases, including thin basement membrane nephropathy (TBMN), Alport syndrome (AS), and late-onset focal segmental glomerulosclerosis (FSGS) [2].

TBMN is characterized by the observation of a thin GBM on kidney biopsy specimens, microhematuria, minimal proteinuria < 500 mg/day, and normal kidney function [3]. TBMN is thought to be present in 1% of the adult population [4]. TBMN is defined as a GBM thickness of <265 nm in an adult patient [5]. The main cause of TBMN is considered to be heterozygous mutations of *COL4A3* or *COL4A4* [6,7,8].

AS is characterized by progressive kidney disease, sensorineural hearing loss, and eye abnormalities [9]. While the causative gene for X-linked AS is *COL4A5*, the causative genes for autosomal recessive AS are *COL4A3* and *COL4A4*. Moreover, heterozygous mutations of *COL4A3* or *COL4A4* are reported to cause autosomal dominant Alport syndrome (ADAS) [10,11]. Although there is currently conjecture as to exactly how to define TBMN or ADAS, with the performance of next generation sequencing, the identification rate of ADAS has increased up to 30% of AS cases [12,13].

FSGS is characterized by sclerosis in focal segmental glomeruli on kidney biopsy specimens, proteinuria, and progressive kidney disease in most cases [14]. Several studies have reported that the patients with heterozygous mutations of *COL4A3* or *COL4A4* developed late-onset FSGS [15,16,17,18].

Since patients with heterozygous mutations of *COL4A3* or *COL4A4* have shown a wide spectrum of disease from TBMN to ADAS, a systematic review has been performed and shown that 29% developed chronic kidney disease (CKD) and 15.1% reached end-stage kidney disease (ESKD) [2]. In the present study, we analyzed the *COL4A3* and *COL4A4* genes in TBMN cases using Sanger sequencing, multiplex ligation-dependent probe amplification (MLPA), and exome sequencing.

## 2. Materials and Methods

### 2.1. Patients

Patients who were being treated in the outpatient ward between July 2019 and July 2021 were enrolled in the present study. The inclusion criteria were as follows: age 20–80 years, presence of microhematuria, undergoing kidney biopsy, and having a GBM thickness of <265 nm [5]. Microhematuria was defined as >5 red blood cells per high power field. In cases in which kidney biopsies were performed at another hospital and the results of electron microscopy were not available, patients whose diagnoses of TBMN were confirmed in another hospital were included. Family history of microhematuria was considered positive if more than one family member had microhematuria. Patients < 20 or > 80 years of age, patients without microhematuria, patients who did not receive a kidney biopsy, patients whose kidney biopsy specimen had a GBM thickness of >265 nm, and patients who did not give their consent were excluded from this study.

After obtaining written informed consent from 13 patients, blood samples were collected.

### 2.2. DNA Extraction

Genomic DNA was extracted from the blood samples of the 13 patients using a Blood Genomic DNA Extraction Mini Kit (FAVORGEN, Shuttleworthstraße, Vienna, Austria). The DNA concentration was measured with a DS-11 Series Spectrophotometer (DeNovix, Wilmington, NC, USA). Each exon of the *COL4A3* or *COL4A4* gene was amplified with each of the primer pairs, which were designed using the Primer3 program (Appendix A) [19].

### 2.3. Sanger Sequencing

Polymerase chain reaction (PCR) was performed using TaKaRa PCR Thermal Cycler Dice Touch (Takara Bio, Shiga, Japan). The PCR conditions were as follows: 96 °C for 15 min, 35 cycles of denaturation at 96 °C for 45 s, annealing at 57 °C for 45 s, and elongation at 72 °C for 1 min, and 72 °C for 15 min. Gel electrophoresis of the PCR products was performed in 1% agarose with Submerge-Mini (Atto, Tokyo, Japan) at 100V for 20 min. If the PCR products had a single band, they were cleaned with ExoSAP-IT PCR product cleanup reagent (Thermo Fisher Scientific, Waltham, MA, USA) at 37 °C for 15 min, followed by incubation at 80 °C for 15 min. If the PCR products had more than two bands with gel electrophoresis, DNA fragments were excised with a clean scalpel, and gel extraction was performed with a Monarch DNA Gel Extraction Kit (New England Biolabs, Ipswich, MA, USA). Sanger sequencing was performed with an Applied Biosystems 3130 Genetic Analyzer (Applied Biosystems, Waltham, MA, USA). The sequence results of each exon or splice site were analyzed with the BioEdit software program and compared using the Ensembl database.

### 2.4. Multiplex Ligation-Dependent Probe Amplification (MLPA) Analyses

MLPA analyses were performed with a SALSA MLPA kit P439 (COL4A3) or SALSA MLPA kit P444 (COL4A4) (MRC Holland, Amsterdam, the Netherlands) to detect deletions or duplications in the *COL4A3* or *COL4A4* genes, respectively.

### 2.5. Exome Sequencing

Samples without any positive results in Sanger sequencing were analyzed by exome sequencing. The Ion AmpliSeq Exome RDY Library (Thermo Fisher Scientific) was prepared according to the manufacturer’s instructions. The library concentration was examined with a QuantStudio 3D digital PCR system (Thermo Fisher Scientific). Templates were prepared with an Ion PI Hi-Q Chef kit (ThermoFisher Scientific) and sequenced with an Ion Proton system (Thermo Fisher Scientific).

### 2.6. Pathogenicity Evaluation

The pathogenicity of the identified variants was examined according to the American College of Medical Genetics and Genomics (ACMG) guidelines [20]. The minor allele frequency (MAF) was defined as common, low, and rare if it was >5%, 1–5%, and <1%, respectively [21]. The pathogenicity of variants was assessed in silico with software programs such as PolyPhen-2 and PROVEAN.

## 3. Results

### 3.1. Background Data

Thirteen patients were analyzed (Table 1). The mean age of the 13 patients was 45 years, and 8 patients were female. There was a family history of microhematuria in two patients. Four patients had hypertension, and two had hyperlipidemia. All patients had microhematuria. The median protein creatinine ratio was 0.15 (0.02–0.62) g/g⋅Cr and the protein creatinine ratio was > 0.15 g/g⋅Cr in six patients; the protein creatinine ratio was > 0.5 g/g⋅Cr in five of these patients. The mean estimated glomerular filtration rate (eGFR) was 80.9 ± 18.9 mL/min/1.73 m^2^ and one patient had an eGFR was <60 mL/min/1.73 m^2^. All patients received kidney biopsy; among these cases, electron microscopy results could be obtained for 10 patients because 3 patients received kidney biopsy in other hospitals. The mean width of the GBM in 10 patients was 208.8 ± 25.1 nm. Sanger sequencing was performed in 13 patients, and variants in *COL4A3* or *COL4A4* were detected in 9 patients.

### 3.2. Mutation Analyses of Thin Basement Membrane Nephropathy

#### 3.2.1. Sanger Sequencing

There were 10 heterozygous variants in *COL4A3* or *COL4A4* in nine patients; among these, eight were single nucleotide variants and two were deletion variants (Figure 1, Table 2). There was a splice-site single nucleotide variant in Patient 6. Patient 1 had digenic variants (*COL4A3* c.469G > C and *COL4A4* c.2510G > C). Seven different glycine substitutions in the collagenous domain were identified in six patients. The pathogenicity of the 10 variants was examined according to the ACMG guidelines [20] and 5, 3, and 2 of the 10 variants were considered to be “likely pathogenic”, of “uncertain significance”, and “pathogenic”, respectively (Table 2 and Appendix A). Three variants (*COL4A4* c.2573G > A in Patient 5, *COL4A3* c.2969_2980del in Patient 7, and *COL4A4* c.904delG in Patient 9) were unreported in the ClinVar database [22] and Leiden Open Variation Database (LOVD) [23] and were considered to be novel variants. The minor allele frequency (MAF) was examined in each single nucleotide polymorphism in the gnomAD and 8.3KJPN databases, and rare variants with “uncertain significance” or “likely pathogenic” were observed in three or one patients, respectively. The three rare variants with “uncertain significance” were “probably damaging” and “deleterious” in PolyPhen-2 and PROVEAN programs, respectively.

#### 3.2.2. MLPA Analyses

MLPA analyses of *COL4A3* and *COL4A4* were performed in four patients with no apparent variants on Sanger sequencing, which yielded negative results (Figure 2).

#### 3.2.3. Exome Sequencing

Exome sequencing was performed for four patients with no apparent variants for Sanger sequencing. Fifty-eight genes that were reported to be causative genes of steroid resistant nephrotic syndrome or modifier genes of TBMN were examined [24,25]. The 58 genes were as follows: *ACTN4, ADCK4, ALG1, ANLN, APOL1, ARHGAP24, ARHGDIA, CD151, CD2AP, CFH, COL4A3, COL4A4, COL4A5, COQ2, COQ6, CRB2, CUBN, DGKE, E2F3, EMP2, FAT1, GPC5, INF2, ITGA3, ITGB4, KANK1, KANK2, KANK4, LAMB2, LMNA, LMX1B, MAGI2, MTTL1, MYH9, MYO1E, NEPH3, NPHS1, NPHS2, NUP107, NUP205, NUP93, NXF5, OCRL1, PAX2, PDSS2, PLCE1, PMM2, PODXL, PTPRO, SCARB2, SMARCAL1, SYNPO, TRPC6, TTC21B, WDR73, WT1, XPO5,* and *ZMPSTE24*. Three heterozygous rare variants were identified (*KANK1* c.2896 + 2T > G in Patient 10, *NPHS1* c.2869G > C in Patient 12, and *NUP205* c.4799T > A in Patient 13) (Appendix A). The pathogenicity of *KANK1* c.2896 + 2T > G was considered to be “pathogenic”, while the pathogenicity of *NPHS1* c.2869G > C and *NUP205* c.4799T > A was considered to be of “uncertain significance” according to the ACMG guidelines [20].

## 4. Discussion

We demonstrated 10 heterozygous variants in *COL4A3* or *COL4A4* in 9 of 13 patients and the detection rate was 69.2%. The diagnostic rate of “likely pathogenic” or “pathogenic” under the ACMG guidelines in the present study was 53.8% (7 out of 13 patients). Three of the 10 variants were novel ones. While MLPA analyses yielded negative results, exome sequencing identified three heterozygous variants in causative genes of FSGS in four patients with no apparent variants on Sanger sequencing.

Of the three novel variants, two were deletion variants. As *COL4A3* c.2969_2980del (p.Ala990_Pro993del) in Patient 7 was an in-frame four-amino-acid deletion, and the deletion included glycine residue at position 991; it was considered to be “likely pathogenic” under the ACMG guidelines. On the other hand, *COL4A4* c.904delG (p.Gly302ValfsTer23) in Patient 9 caused a premature stop codon and was considered to be “pathogenic” under the ACMG guidelines. While Patient 7 had a preserved kidney function without any comorbidities, Patient 9 had proteinuria with obesity and required treatment with an angiotensin II receptor blocker (ARB) and a sodium glucose cotransporter 2 (SGLT2) inhibitor. One of the three novel variants *COL4A4* c.2573G > A (p.Gly858Glu) which was not reported in the ClinVar or LOVD databases was in Patient 5, while *COL4A4* c.2573G > C was reported in a previous study [12]. The pathogenicity was considered to be “likely pathogenic” under the ACMG guidelines. Since the number of side chain carbon atoms was associated with the age of onset of ESKD in a patient with glycine substitution [26], regular follow-up will be necessary for Patient 5.

Digenic variants (*COL4A3* c.469G > C and *COL4A4* c.2510G > C) were identified in Patient 1. *COL4A3* c.469G > C (p.Gly157Arg) was in front of the first interruption of α3 (IV) [27] and *COL4A4* c.2510G > C (p.Gly837Ala) was reported to be pathogenic [28]. Previous reports showed that the progression of kidney dysfunction in patients with two pathogenic mutations in different genes could be intermediate between ADAS and autosomal recessive AS [29]. Since the renoprotective effects of angiotensin-converting enzyme (ACE) inhibitors before overt hypertension in AS were shown [30], renin–angiotensin system (RAS) inhibitor treatment should be initiated even if the patient was not hypertensive.

Four known variants, *COL4A3* c.697G > A (p.Gly233Arg), *COL4A3* c.1229G > A (p.Gly410Glu), *COL4A4* c.827G > C (p.Gly276Ala), and *COL4A4* c.1733G > T (p.Gly578Val), were also identified in Patients 2, 3, 4, and 8, respectively, in the present study. As Patients 2 and 3 had hypertension and a protein creatinine ratio of > 0.5 g/g⋅Cr, Patient 2 was treated with an ARB, a calcium channel blocker (CCB), and an SGLT2 inhibitor, while Patient 3 was treated with an ARB and a CCB.

A known splice-site single nucleotide variant (*COL4A4* c.71 + 1G > A) was identified in Patient 6. He was carefully followed up without medication because his kidney function was preserved without proteinuria.

Regarding the follow-up testing of family members, only the little sister of the Patient 2, who had microhematuria, received a genetic analysis. The same heterozygous variant as Patient 2 was identified. Although Patient 1 had digenic variants, genetic analyses of family members could not be performed because we could not obtain their consent. The mother of Patient 8 was diagnosed with TBMN without a genetic evaluation based on a kidney biopsy, and her maternal grandfather died of ESKD. We could not perform genetic analyses for other family members.

In a systematic review of 777 patients with heterozygous *COL4A3/COL4A4* mutations in 258 families, 28.6% of whom received kidney biopsy; microhematuria with/without macrohematuria, proteinuria (>0.5 g/day), CKD, and ESKD were observed in 89.1%, 41.6%, 29%, and 15.1%, respectively [2]. Non-missense mutations were detected in 27% of 74 patients who developed ESKD [2]. As far as we know, one article with heterozygous *COL4A3/COL4A4* mutations has been identified after the publication of a systematic review [31]. In a report of 53 patients with heterozygous *COL4A3/COL4A4* mutations in 25 families, microhematuria, proteinuria (>0.2 g/g⋅Cr), CKD, and ESKD were observed in 100%, 17%, 7.5%, and 5.7% of subjects, respectively [31]. All three patients who developed ESKD had missense mutations [31]. In the present study, there were eight patients with heterozygous *COL4A3/COL4A4* variants, excluding Patient 1, who showed digenic inheritance. Microhematuria and proteinuria (>0.5 g/g⋅Cr) were observed in 100% and 37.5% of patients, respectively. From the two articles and an article from Yang C et al., we collected 145 patients with heterozygous *COL4A3/COL4A4* mutations and microhematuria whose diagnoses were confirmed by a kidney biopsy [2,31,32], Appendix A. The numbers of patients with missense mutations were higher than those with non-missense mutations in ADAS, TBMN, TBMN-FSGS and than in the present study (Table 3). The prevalence of CKD or proteinuria (>0.5g/day) was higher in ADAS than in TBMN or TBMN-FSGS and the prevalence in the present study fell between that in ADAS and TBMN or TBMN-FSGS (Table 3). Regarding missense mutations, the prevalence of CKD, proteinuria (>0.5 g/day), or ESKD cases in patients with glycine substitution was higher than that in patients with non-glycine substitution in the 3 articles, and the prevalence of CKD or proteinuria (>0.5 g/day) in patients with glycine substitution in the present study also fell between that in patients with glycine substitution in ADAS and TBMN or TBMN-FSGS (Appendix A). Regarding non-missense mutations, the prevalence of proteinuria (>0.5 g/day) in patients with deletion in the present study fell between that in patients with deletion in ADAS and TBMN or TBMN-FSGS (Appendix A). Although long follow-up was not conducted in the present study, previous reports showed that patients with *COL4A4* c.827G > C or *COL4A4* c.1733G > T developed ESKD at 58 or 63 years of age, respectively [29,33]; therefore, careful follow-up is necessary for Patients 4 and 8. Three (Patients 6, 7, and 9) out of eight variants (37.5%) had a non-missense mutation in the present study. As the age at the onset of ESKD was younger in patients with non-missense mutations in comparison to those with missense mutations [2], the three patients in the present study should be followed up carefully.

Exome sequencing in four patients with no apparent variants on Sanger sequencing identified three heterozygous variants in causative genes of FSGS and the pathogenicity of *KANK1* c.2896 + 2T > G was “pathogenic” according to the ACMG guidelines, Patient 10 has been treated with an ACE inhibitor and an SGLT2 inhibitor.

The present study was associated with the following limitations. In the present study, exome sequencing plus MLPA analyses might have been a better approach than Sanger sequencing plus MLPA analyses. As the costs for Sanger sequencing were lower than exome sequencing once primer sets were purchased, we chose Sanger sequencing—a more laborious process—instead of exome sequencing. MLPA analyses showed negative results, which might be compatible with the rarity of large deletion or duplication. Patient 10 was diagnosed with IgA nephropathy, and thin GBM might be attributed to this disease. Finally, we did not examine variants in the introns of *COL4A3* or *COL4A4* in the present study.

## 5. Conclusions

In conclusion, careful follow-up will be necessary for patients with heterozygous variants of *COL4A3* or *COL4A4*, since they showed a wide spectrum of disease, ranging from TBMN to ADAS.

## Figures and Tables

**Figure 1 genes-13-01779-f001:**
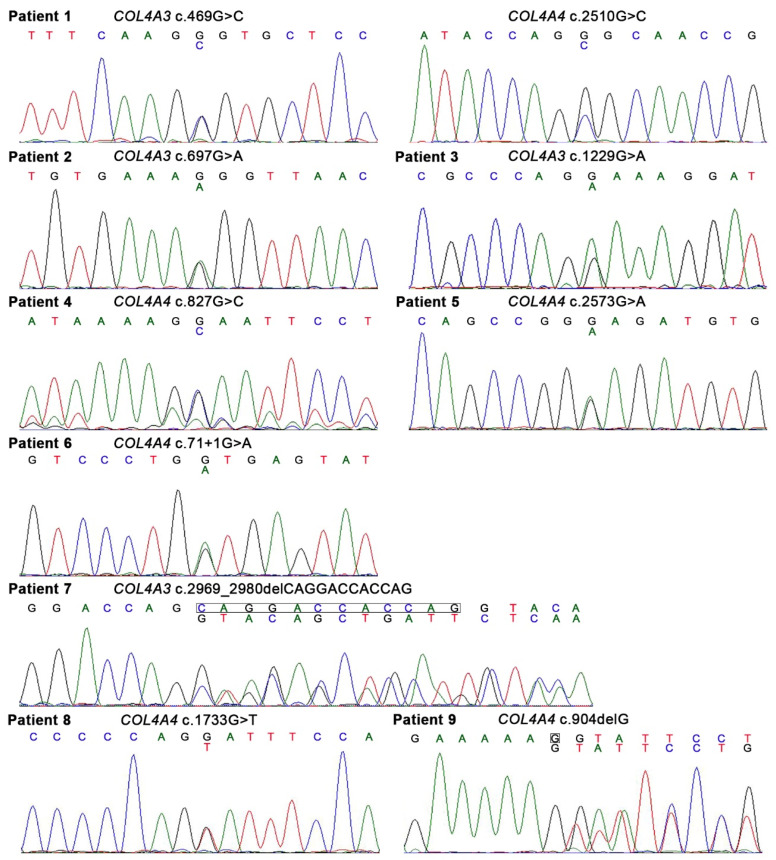
Heterozygous variants of *COL4A3* or *COL4A4* identified in the present study. Of 10 variants in *COL4A3* or *COL4A4*, 8 were single nucleotide variants and 2 were deletion variants. There was a splice-site single nucleotide variant in Patient 6. Patient 1 had digenic variants (*COL4A3* c.469G > C and *COL4A4* c.2510G > C).

**Figure 2 genes-13-01779-f002:**
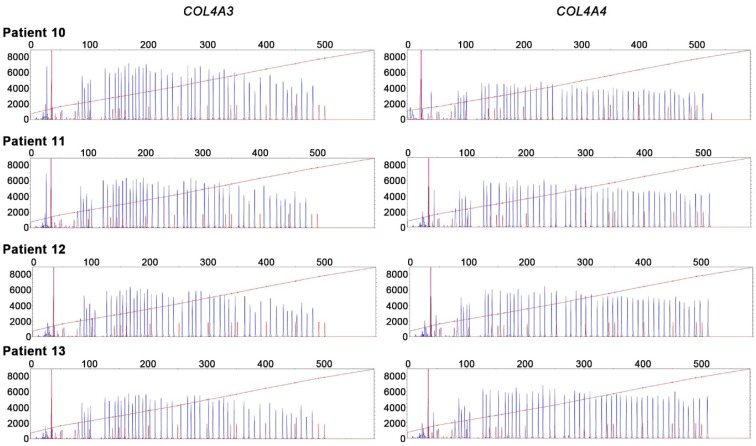
Multiplex ligation-dependent probe amplification (MLPA) analyses of *COL4A3* and *COL4A4*. MLPA analyses showed negative results in four patients with no apparent variants for Sanger sequencing.

**Table 1 genes-13-01779-t001:** Clinical characteristics of the study subjects.

	Age	Sex	Family History	Comorbidity	Microhematuria (/HF)	Proteinuria (g/g Cr)	eGFR (mL/min/1.73 m^2^)	Biopsy	GBM (nm)	Variants
*Sanger*										
Pt 1	37	M	(−)	(−)	30 to 49	0.34	88.5	(+)	NA	(+)
Pt 2	60	F	(+)	HT	20 to 29	0.54	55.7	(+)	210.7 ± 57.1	(+)
Pt 3	44	F	(−)	HT	50 to 99	1.1	105.6	(+)	177.9 ± 35.7	(+)
Pt 4	41	F	(−)	HL	10 to 19	0.01	86.4	(+)	195.2 ± 57.5	(+)
Pt 5	22	M	(−)	(−)	50 to 99	0.02	121.8	(+)	232.8 ± 40.1	(+)
Pt 6	36	M	(−)	(−)	50 to 99	0.01	87.3	(+)	NA	(+)
Pt 7	36	F	(−)	(−)	50 to 99	0.01	87.3	(+)	181.7 ± 37.1	(+)
Pt 8	40	F	(+)	(−)	30 to 49	0.15	74.6	(+)	NA	(+)
Pt 9	49	F	(−)	Obesity	30 to 49	0.81	60.7	(+)	174.4 ± 62.1	(+)
*Exome*										
Pt 10	64	F	(−)	IgA-N	10 to 19	0.88	70.8	(+)	215.4 ± 58.0	(−)
Pt 11	50	M	(−)	HT	5 to 9	0.05	60.1	(+)	242.1 ± 54.7	(−)
Pt 12	69	F	(−)	HT, HL	30 to 49	0.09	67	(+)	223.7 ± 43.5	(−)
Pt 13	41	M	(−)	(−)	5 to 9	0.62	85.3	(+)	234.2 ± 47.3	(−)

Data indicate the mean ± standard deviation. eGFR, estimated glomerular filtration rate; F, female; GBM, glomerular basement membrane; HF, high power field; HL, hyperlipidemia; HT, hypertension; IgA-N, IgA nephropathy; M, male; NA, not available; Pt, patient.

**Table 2 genes-13-01779-t002:** Summary of the mutation analyses.

	Sanger Sequencing Results	Amino Acid Change	ACMG	ClinVar	LOVD	dbSNP	gnomAD	8.3KJPN	Variant Category	PolyPhen-2 (Score)	PROVEAN (Score)
Pt 1	*COL4A3* c.469G > C	p.Gly157Arg	US	US	Unreported	rs764451365	0.000016	ND	Rare	PD (1.00)	Deleterious (−6.778)
Pt 1	*COL4A4* c.2510G > C	p.Gly837Ala	Likely Pathogenic	Pathogenic	Pathogenic	rs201648982	0.000008	0.0009	Rare	PD (1.00)	Deleterious (−5.934)
Pt 2	*COL4A3* c.697G > A	p.Gly233Arg	Likely Pathogenic	Likely Pathogenic	Unreported	ND	ND	ND	ND	PD (1.00)	Deleterious (−6.007)
Pt 3	*COL4A3* c.1229G > A	p.Gly410Glu	US	US	Unreported	rs1350342816	0.000004	ND	Rare	PD (1.00)	Deleterious (−5.457)
Pt 4	*COL4A4* c.827G > C	p.Gly276Ala	US	Unreported	US	rs202242354	0.000016	0.0016	Rare	PD (1.00)	Deleterious (−4.493)
Pt 5	*COL4A4* c.2573G > A	p.Gly858Glu	Likely Pathogenic	Unreported	Unreported	CM148120	ND	ND	ND	PD (1.00)	Deleterious (−7.583)
Pt 6	*COL4A4* c.71 + 1G > A	p.?	Pathogenic	Pathogenic	Likely Pathogenic	rs1559742015	ND	ND	ND	ND	ND
Pt 7	*COL4A3* c.2969_2980del	p.Ala990_Pro993del	Likely Pathogenic	Unreported	Unreported	ND	ND	ND	ND	ND	Deleterious (−9.159)
Pt 8	*COL4A4* c.1733G > T	p.Gly578Val	Likely Pathogenic	Unreported	Pathogenic	CM143835	ND	ND	ND	PD (1.00)	Deleterious (−8.424)
Pt 9	*COL4A4* c.904delG	p.Gly302ValfsTer23	Pathogenic	Unreported	Unreported	ND	ND	ND	ND	ND	ND

ACMG, American College of Medical Genetics and Genomics; del, deletion; dbSNP, Single Nucleotide Polymorphism Database; gnomAD, Genome Aggregation Database; LOVD, Leiden Open Variation Database; ND, no data; PD, probably damaging; PolyPhen-2, Polymorphism Phenotyping v2; PROVEAN, Protein Variation Effect Analyzer; Pt, patient; US, uncertain significance.

**Table 3 genes-13-01779-t003:** Summary of patients with heterozygous *COL4A3/COL4A4* mutations who received a kidney biopsy.

			CKD			Proteinuria			Kidney Outcomes
			eGFR < 60, (%)	eGFR > 60, (%)	ND, (%)	>0.5 g, (%)	<0.5 g, (%)	ND, (%)	ESKD cases, (%)
ADAS	Missense	48	15 (31.3)	22 (45.8)	11 (22.9)	26 (54.2)	14 (29.2)	8 (16.7)	15 (31.3)
	Non-missense	19	4 (21.1)	7 (36.8)	8 (42.1)	13 (68.4)	3 (15.8)	3 (15.8)	4 (21.1)
TBMN	Missense	37	3 (8.1)	16 (43.2)	18 (48.6)	12 (32.4)	13 (35.1)	12 (32.4)	2 (5.4)
	Non-missense	19	1 (5.3)	8 (42.1)	10 (52.6)	6 (31.6)	6 (31.6)	7 (36.8)	1 (5.3)
TBMN-FSGS	Missense	14	1 (7.1)	0 (0.0)	13 (92.9)	3 (21.4)	0 (0.0)	11 (78.6)	9 (64.3)
	Non-missense	7	0 (0.0)	3 (42.9)	4 (57.1)	3 (42.9)	1 (14.3)	3 (42.9)	2 (28.6)
The present study (TBMN)	Missense	5	1 (20)	4 (80)	0 (0.0)	2 (40)	3 (60)	0 (0.0)	0 (0.0)
	Non-missense	3	0 (0.0)	3 (100)	0 (0.0)	1 (33.3)	2 (66.7)	0 (0.0)	0 (0.0)

ADAS, autosomal dominant Alport syndrome; CKD, chronic kidney disease; eGFR, estimated glomerular filtration rate; ESKD, end-stage kidney disease; FSGS, focal segmental glomerulosclerosis; ND, no data; TBMN, thin basement membrane nephropathy.

## Data Availability

The data presented in this study are available on request from the corresponding author. The data are not publicly available due to ethical restrictions.

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
