# Peer review of "Mutation Analysis of Thin Basement Membrane Nephropathy"

_genes, 2022, doi:10.3390/genes13101779_

Round 1

Reviewer 1 Report

Thank you to the authors for submitting this manuscript describing a cohort of 14 cases of TBMN in which genetic analysis has been undertaken. I have some major queries:

1) There is a need for major revision of english syntax and expression throughout.

2) Line 14-16, TBMN is also characterised by a thin GBM on kidney biopsy as a gold standard clinical diagnosis. 

3) Line 18-19, this statement makes no sense.

4) The use of the term "mutation" is erroneous. Please replace this with "variant" unless you are referring to an ACMG classified LP/Path variant in which case "mutation" can be used.

5) The denominator in this study is confusing and absolutely requires correction. 14 patients were included, however 2 of these were related (siblings). I would strongly suggest that 13 probands from 13 unrelated families should be the denominator throughout.

6) Line 39-40, this does not make sense. Please correct.

7) Line 43, "the development stage" requires revision as it is a bizarre term to use

8) the use of tense (past, present, future) and singular/plural requires review and revision throughout

9) Line 53, the rate of ADAS here is very adventurous and the statement does not reflect other studies indicating a lower frequency. There is conjecture at the present as to exactly how to define ADAS vs TBMN vs COL4A-related FSGS.

10) Line 59-62, these statements are confusing and need revision.

11) The description of the included patients in Methods is not clear and is rather confusing.

12) Please move Table 1 to a supplementary

13) Line 116-120, the only key assessment here of importance is the ACMG classification. Others can be used to explore variants, but only ACMG can be used to assort any clinical relevance.

14) Your results require substantial revision to clarify what you are actually trying to say. At the moment it is disorganised and difficult to follow.

15) Table 3, Please replace "mutation with variant" and then add a column as to whether a genetic diagnosis (ACMG LP/Path variant) was identified

16) Remove either patient 2 or patient 10 as these two are related and should not be both considered probands.

17) Table 4, the ONLY column here that is of clear relevance to a genetic outcomes is the ACMG column, so it should be first. Also, please list all ACMG criteria for each individual variant that cumulatively justify the ACMG variant classification.

18) Results - exome Sequencing. This paragraph seems pointless. Please restructure and revise to indicate what you actually want to say. Were all genes analysed? Did you have any candidate genes that you especially looked for??

19) Results - as mentioned before, please use only ACMG LP/Path variants as representing a genetic outcome and also reclarify that your denominator is only actually 13. 

20) Please review your discussion so that key take-home messages are reflected rather than doing a case-by-case walk through of data that is in effect represented (or is better represented) in tables/figures. 

Author Response

To Reviewer 1

1) There is a need for major revision of english syntax and expression throughout.

Response

According to the advice, the manuscript was checked by a professional editor who is a native speaker of English.

2) Line 14-16, TBMN is also characterised by a thin GBM on kidney biopsy as a gold standard clinical diagnosis.

Response

According to the advice, we have changed the sentence to “Thin basement membrane nephropathy (TBMN) is characterized by the observation of microhematuria and a thin glomerular basement membrane on kidney biopsy specimens.”

3) Line 18-19, this statement makes no sense.

Response

According to the advice, we have deleted the sentence.

4) The use of the term "mutation" is erroneous. Please replace this with "variant" unless you are referring to an ACMG classified LP/Path variant in which case "mutation" can be used.

Response

According to the advice, we have replaced “mutation” with “variant” unless we are referring to an ACMG classified LP/Path variant.

5) The denominator in this study is confusing and absolutely requires correction. 14 patients were included, however 2 of these were related (siblings). I would strongly suggest that 13 probands from 13 unrelated families should be the denominator throughout.

Response

According to the advice, we have deleted patient 10. Accordingly, patients 11-14 were changed to patients 10-13, respectively.

6) Line 39-40, this does not make sense. Please correct.

Response

According to the advice, we have changed this to: “TBMN was defined as a GBM thickness of <265 nm in an adult patient.”

7) Line 43, "the development stage" requires revision as it is a bizarre term to use

Response

According to the advice, we have changed this to “a developmental isoform switch from triple-helical α1.α1.α2 (IV) to triple-helical α3.α4.α5 (IV) protomers occurs in the capillary loop stages.”

8) the use of tense (past, present, future) and singular/plural requires review and revision throughout

Response

According to the advice, we have corrected the use of tense (past, present, future) and singular/plural. The manuscript was checked by a professional editor who is a native speaker of English.

9) Line 53, the rate of ADAS here is very adventurous and the statement does not reflect other studies indicating a lower frequency. There is conjecture at the present as to exactly how to define ADAS vs TBMN vs COL4A-related FSGS.

Response

According to the advice, we have changed this to “Although there is currently conjecture as to exactly how to define TBMN or ADAS, with the performance of next generation sequencing, the identification rate of ADAS has increased to up to 30% of AS cases” in the Introduction section on page 2, lines 53-56.

10) Line 59-62, these statements are confusing and need revision.

Response

According to the advice, we have changed this to “In the present study, we analyzed the COL4A3 and COL4A4 genes in TBMN cases by Sanger sequencing, multiplex ligation-dependent probe amplification (MLPA), and exome sequencing.” In the Introduction section on page 2, lines 66-68.

11) The description of the included patients in Methods is not clear and is rather confusing.

According to the advice, we have added patient inclusion and exclusion criteria in the Method section on page 2, lines 71-81.

12) Please move Table 1 to a supplementary

Response

According to the advice, we changed Tables 1, 2, 3, 4 to Tables S1, S2, 1, 2, respectively.

13) Line 116-120, the only key assessment here of importance is the ACMG classification. Others can be used to explore variants, but only ACMG can be used to assort any clinical relevance.

Response

According to the advice, we have added “The pathogenicity of the identified variants was examined according to the American College of Medical Genetics and Genomics (ACMG) guidelines” in the Materials and Methods section on page 3, lines 126-128.

14) Your results require substantial revision to clarify what you are actually trying to say. At the moment it is disorganised and difficult to follow.

Response

According to the advice, we have changed the descriptions in the Results and Discussion sections extensively.

15) Table 3, Please replace "mutation with variant" and then add a column as to whether a genetic diagnosis (ACMG LP/Path variant) was identified

Response

According to the advice, we have replaced “Mutations” with “Variants” in Table 1. As patient 1 had digenic variants and the assessment of the pathogenicity of both variants was necessary. We have summarized the results of the mutation analyses in Table 2.

16) Remove either patient 2 or patient 10 as these two are related and should not be both considered probands.

Response

According to the advice, we have deleted patient 10.

17) Table 4, the ONLY column here that is of clear relevance to a genetic outcomes is the ACMG column, so it should be first. Also, please list all ACMG criteria for each individual variant that cumulatively justify the ACMG variant classification.

Response

According to the advice, we have moved the ACMG column next to the “Amino acid change” column. We have shown all of the ACMG criteria in Supplementary Table S3.

18) Results - exome Sequencing. This paragraph seems pointless. Please restructure and revise to indicate what you actually want to say. Were all genes analysed? Did you have any candidate genes that you especially looked for??

Response

According to the advice, we have changed the paragraph and added “Fifty-eight genes that were reported to be causative genes of steroid resistant nephrotic syndrome or modifier genes of TBMN were examined. The 58 genes were as follows: ACTN4, ADCK4, ALG1, ANLN, APOL1, ARHGAP24, ARHGDIA, CD151, CD2AP, CFH, COL4A3, COL4A4, COL4A5, COQ2, COQ6, CRB2, CUBN, DGKE, E2F3, EMP2, FAT1, GPC5, INF2, ITGA3, ITGB4, KANK1, KANK2, KANK4, LAMB2, LMNA, LMX1B, MAGI2, MTTL1, MYH9, MYO1E, NEPH3, NPHS1, NPHS2, NUP107, NUP205, NUP93, NXF5, OCRL1, PAX2, PDSS2, PLCE1, PMM2, PODXL, PTPRO, SCARB2, SMARCAL1, SYNPO, TRPC6, TTC21B, WDR73, WT1, XPO5, ZMPSTE24. Three heterozygous rare variants were identified (KANK1 c.2896+2T>G in patient 10, NPHS1 c.2869G>C in patient 12, and NUP205 c.4799T>A in patient 13) (Table S4). The pathogenicity of KANK1 c.2896+2T>G was considered to be “pathogenic”, while the pathogenicity of NPHS1 c.2869G>C and NUP205 c.4799T>A was considered to be of “uncertain significance” according to the ACMG guidelines [20].”

19) Results - as mentioned before, please use only ACMG LP/Path variants as representing a genetic outcome and also reclarify that your denominator is only actually 13.

Response

According to the advice, we have changed the Results section and removed patient 10.

20) Please review your discussion so that key take-home messages are reflected rather than doing a case-by-case walk through of data that is in effect represented (or is better represented) in tables/figures.

Response

According to the advice, we have changed the Discussion section extensively.

Reviewer 2 Report

In their manuscript, ‘Mutation analysis of thin basement membrane nephropathy,’ Hirabayashi and colleagues present an analysis of the clinical and pathologic features of fourteen patients with thin basement nephropathy (TBMN). They report that they identified heterozygous COL4A3 or COL4A4 variants in ten of the fourteen (71.4%) patients, and that these individuals encompassed a broad clinical spectrum of disease. Overall, the manuscript is well-organized and grammatically written. The phenotypic spectrum associated with heterozygous COL4A3 and COL4A4 variants is an area of interest to the nephrology community, especially clinicians and researchers in the field of nephrogenetics. However, given that others have reported on this topic in larger cohorts (e.g., PMID: 35090027, PMID: 33391746), it is unclear what novel insight the study provides. Therefore, the manuscript would be greatly strengthened by incorporating additional analyses, e.g., a systematic comparison of the findings in their cohort with the results of previously published genetic studies of individuals with TBMN.

Author Response

To Reviewer 2

In their manuscript, ‘Mutation analysis of thin basement membrane nephropathy,’ Hirabayashi and colleagues present an analysis of the clinical and pathologic features of fourteen patients with thin basement nephropathy (TBMN). They report that they identified heterozygous COL4A3 or COL4A4 variants in ten of the fourteen (71.4%) patients, and that these individuals encompassed a broad clinical spectrum of disease. Overall, the manuscript is well-organized and grammatically written. The phenotypic spectrum associated with heterozygous COL4A3 and COL4A4 variants is an area of interest to the nephrology community, especially clinicians and researchers in the field of nephrogenetics. However, given that others have reported on this topic in larger cohorts (e.g., PMID: 35090027, PMID: 33391746), it is unclear what novel insight the study provides. Therefore, the manuscript would be greatly strengthened by incorporating additional analyses, e.g., a systematic comparison of the findings in their cohort with the results of previously published genetic studies of individuals with TBMN.

Response

According to the advice, we have compared our findings with two cohorts (PMID: 35090027 and PMID: 33391746). We have added “In a systematic review of 777 patients with heterozygous COL4A3/COL4A4 mutations in 258 families, 28.6% of whom received kidney biopsy, microhematuria with/without macrohematuria, proteinuria (>0.5 g/day), CKD, and ESKD were observed in 89.1%, 41.6%, 29%, and 15.1%, respectively [2]. Non-missense mutations were detected in 27% of 74 patients who developed ESKD [2]. In a report of 53 patients with heterozygous COL4A3/COL4A4 mutations in 25 families, microhematuria, proteinuria (>0.2 g/g·Cr), CKD, and ESKD were observed in 100%, 17%, 7.5%, and 5.7% of patients, respectively [31]. All three patients who developed ESKD had missense mutations [31]. In the present study, there were 8 patients with heterozygous COL4A3/COL4A4 variants, excluding patient 1 who showed digenic inheritance. Microhematuria and proteinuria (>0.5 g/g·Cr) were observed in 100% and 37.5% of patients, respectively. Although long follow-up was not conducted in the present study, previous reports showed that patients with COL4A4 c.827G>C or COL4A4 c.1733G>T developed ESKD at 58 or 63 years of age, respectively [29, 32]; therefore, careful follow-up is needed for patients 4 and 8. Three (patient 6, 7, and 9) out of 8 variants (37.5%) had a non-missense mutation in the present study. As the age at the onset of ESKD was younger in patients with non-missense mutations in comparison to those with missense mutations [2], the three patients in the present study should be followed carefully.” in the Discussion section.

Round 2

Reviewer 1 Report

Thank you for the revisions. The manuscript is improved, however the results and abstract remain very confusing. The diagnostic rate is unclear. The authors persist with a very loose approach to describing that variants are present without defining how many of the participants secured a genetic diagnosis (an LP/P variant according to ACMG criteria). The reporting of the results is thus confusing in parts and difficult to engage with.

Author Response

As suggested, we have added, "The diagnostic rate of "likely pathogenic" or "pathogenic" under the American College of Medical Genetics and Genomics guidelines was 53.8% (7 out of 13 patients)." to the Abstract section. We have also added, "The diagnostic rate of "likely pathogenic" or "pathogenic" under the ACMG guidelines in the present study was 53.8% (7 out of 13 patients)." to the Discussion section.

Reviewer 2 Report

I appreciate the authors revising per my comments and the comments of the other reviewers. While the revised manuscript is improved versus the original, the novelty of its findings remains unclear given that, as mentioned in my comments and those of the other reviewers, multiple larger-scale analyses of individuals with heterozygous type IV collagen variants have been performed. Additional analysis/analyses would be helpful to address this issue. 

Author Response

As suggested, we collected patients with heterozygous COL4A3/COL4A4 mutations and microhematuria whose diagnoses were confirmed by a kidney biopsy from the previous two articles and summarized the results as Table 3 and Table S5. We have now mentioned this in the Discussion section as follows:

"From the previous 2 articles, we collected 144 patients with heterozygous COL4A3/COL4A4 mutations and microhematuria whose diagnoses were confirmed by a kidney biopsy [2, 31, Table S5]. The numbers of patients with missense mutations were higher than those with non-missense mutations in ADAS, TBMN, TBMN-FSGS, and the present study [Table 3]. The prevalence of CKD or proteinuria (>0.5 g/day) was higher in ADAS than in TBMN or TBMN-FSGS, and the prevalence in the present study fell between that in ADAS and TBMN or TBMN-FSGS [Table 3]."

Round 3

Reviewer 2 Report

I appreciate Hirabayashi and colleagues revising their manuscript further per my comments and the comments of the other reviewers. The addition of a more detailed comparison of the cases reported in their manuscript to the cases previously reported in the published literature (see Table S2) helps increase the significance of their study. I would suggest expanding the associated section about this comparison in the Discussion further, including a more detailed analysis of genotype-phenotype correlations - e.g., do the clinical features of individuals with missense variants that are glycine substitutions differ from those with missense variants that are not glycine substitutions?

Author Response

As suggested, we conducted more detailed analyses and have now added the following to the Discussion section: “Regarding missense mutations, the prevalence of CKD, proteinuria (>0.5 g/day) or ESKD cases in patients with glycine substitution was higher than that in patients with non-glycine substitution in the 2 articles, and the prevalence of CKD or proteinuria (>0.5 g/day) in patients with glycine substitution in the present study also fell between that in patients with glycine substitution in ADAS and TBMN or TBMN-FSGS [Table S5]. Regarding non-missense mutations, the prevalence of proteinuria (>0.5 g/day) in patients with deletion in the present study fell between that in patients with deletion in ADAS and TBMN or TBMN-FSGS [Table S5].”
